# Clinical and economic burden of low back pain in low- and middle-income countries: a systematic review

Francis Fatoye [ID],[1,2] Tadesse Gebrye,[1] Chidozie Emmanuel Mbada,[3] Ushotanefe Useh[2]

[1]Department of Health Professions, Manchester Metropolitan University, Manchester, UK
[2]Lifestyle Diseases, Faculty of Health Sciences, North West University, Mmabatho, South Africa
[3]Department of Medical Rehabilitation, Obafemi Awolowo University, Ile-Ife, Nigeria

**Correspondence to**
Professor Francis Fatoye;
f.fatoye@mmu.ac.uk

## ABSTRACT

**Objectives** Low back pain (LBP) is the leading cause of disability and work absenteeism globally, and it poses significant clinical and economic burden to individuals, health systems and the society. This study aimed to synthesise the clinical and economic burden of LBP in low-income and middle-income countries (LMICs).

**Methods** A systematic review following the Preferred Reporting Items for Systematic Reviews and Meta-Analyses guidelines was performed. PubMed, Medline, CINAHL, PsycINFO, AMED, Embase and Scopus databases were systematically searched for studies that examined the clinical and economic burden of LBP in LMICs, published from inception to 10 December 2021. Only studies with clearly stated methodologies and published in English were eligible for review.

**Results** Nine studies met the inclusion criteria and were reviewed. Of these, three of them were clinical burden studies. The mean Newcastle–Ottawa Quality Assessment Scale (NOS) score of the included studies was 4, with an average from 3 to 6. The included studies were conducted in Argentina, Brazil, China, Ethiopia, Nigeria and Republic of Serbia. The rates of hospitalisation due to LBP ranged between 13.4% and 18.7%. Due to variation of methodological approaches, the reported cost estimates were inconsistent across the studies. A total cost of US\$2.2 billion per population and US\$1226.25 per patient were reported annually due to LBP.

**Conclusion** This systematic literature review suggests that LBP is associated with significantly high rates of hospitalisation and costs. As LBP is an important threat to the population, health professionals and policymakers are to put in place appropriate programmes to reduce the clinical and economic burden associated with LBP and improve the health outcomes of individuals with this condition in LMICs.

**PROSPERO registration number** CRD42020196335.

## INTRODUCTION

Low-back pain (LBP) is an important health problem and covers radicular, axial lumbosacral and referred pain.[1] The lifetime, 1 year and point prevalence of LBP worldwide ranged from 11% to 84%, 22% to 65% and 12% to 33%, respectively.[2] The 1-year incidence of people who have a first-ever episode of LBP ranged from 6.5% to 15.4% and 1.5% to 36%.[3] LBP is associated with increased healthcare costs, length of stay in hospitals, morbidity and mortality in both high-income and low-income countries.[4]

To reduce the clinical and economic burden associated with LBP, it is important that prevention and management strategies of LBP are implemented. Prevention and management strategies are aimed at controlling pain, maintaining function and prevention of exacerbation of LBP.[5] The American College of Physicians and the American Pain Society recommend that individual with LBP should remain active, and they provide information about effective self-care options, use of medications with proven benefits, interdisciplinary rehabilitation, exercise, acupuncture, massage, spinal manipulation, yoga, cognitive behavioural therapy or relaxation for the treatment of LBP.[6]

The implementation of these treatments remains a challenge for clinical practice and research.[7] Because of this challenge, LBP was listed among the top 10 diseases and injuries that account for the highest number of disability-adjusted life years worldwide.[8] The direct medical and indirect costs of LBP are more than US\$50 billion per annum and could be as high as US\$100 billion at the extreme.[9] Evidence suggests that as much as

### STRENGTHS AND LIMITATIONS OF THIS STUDY

⇒ This is the first study that summarises and critically appraises the current evidence base on clinical and economic impact of low back pain in low-income and middle-income countries and discusses the potential avenues for future research.
⇒ This review covers a wide variety of the literature and follows the Preferred Reporting Items for Systematic Reviews and Meta-Analyses.
⇒ The included literature were small and failed to fully report their data which impacted the quality ratings and the content of the meta-analyses.

80%–90% of the populations' work in low-income and middle-income countries (LMICs) entails heavy labour such as carrying heavy loads on the back or head.[10] The risk factors of LBP include low educational status, stress, anxiety, depression, job dissatisfaction, low levels of social support in the workplace and whole-body vibration.[3] The causes and risk factors for LBP in LMIC is substantial.[11] Thus, the likelihood of having LBP is higher in LMICs than in high-income countries.

Many studies have been published evaluating the clinical and economic burden of LBP in LMICs. However, to date there are no studies that have summarised literature on the clinical and economic burden of LBP in LMICs. Therefore, this systematic review critically evaluated and summarised the results of all available published systematic reviews that have investigated the clinical and economic burden of LBP in LMICs.

## METHODS
### Search protocol and registration
In this study, we used the Preferred Reporting Items for Systematic Reviews and Meta-Analysis guideline. A protocol for this systematic review was prospectively registered on PROSPERO and can be found at https://www.crd.york.ac.uk/prospero/#recordDetails, ID=CRD42020196335.

### Search strategy
The systematic review was carried out in the PubMed, Medline, CINAHL, PsycINFO, AMED, Embase and Scopus databases with studies published from inception to 10 December 2021. The following keywords were used in the search: low back pain, hospitalisation, cost of illness, absenteeism, ambulatory care, drug costs, emergency medical services, healthcare costs, nursing services, economics, physicians visit, clinical impact, utilisation, burden of illness, cost, nursing cost (online supplemental appendix 1). These search terms were combined using conjunctions words 'AND' or 'OR'. Further, a manual search of reference sections of the included studies was also checked for additional studies. The search was performed independently by two authors (TG and FF) to avoid the presence of bias in the selection and exclusion of studies. Any disagreement was resolved by discussion with the third the author (CEM).

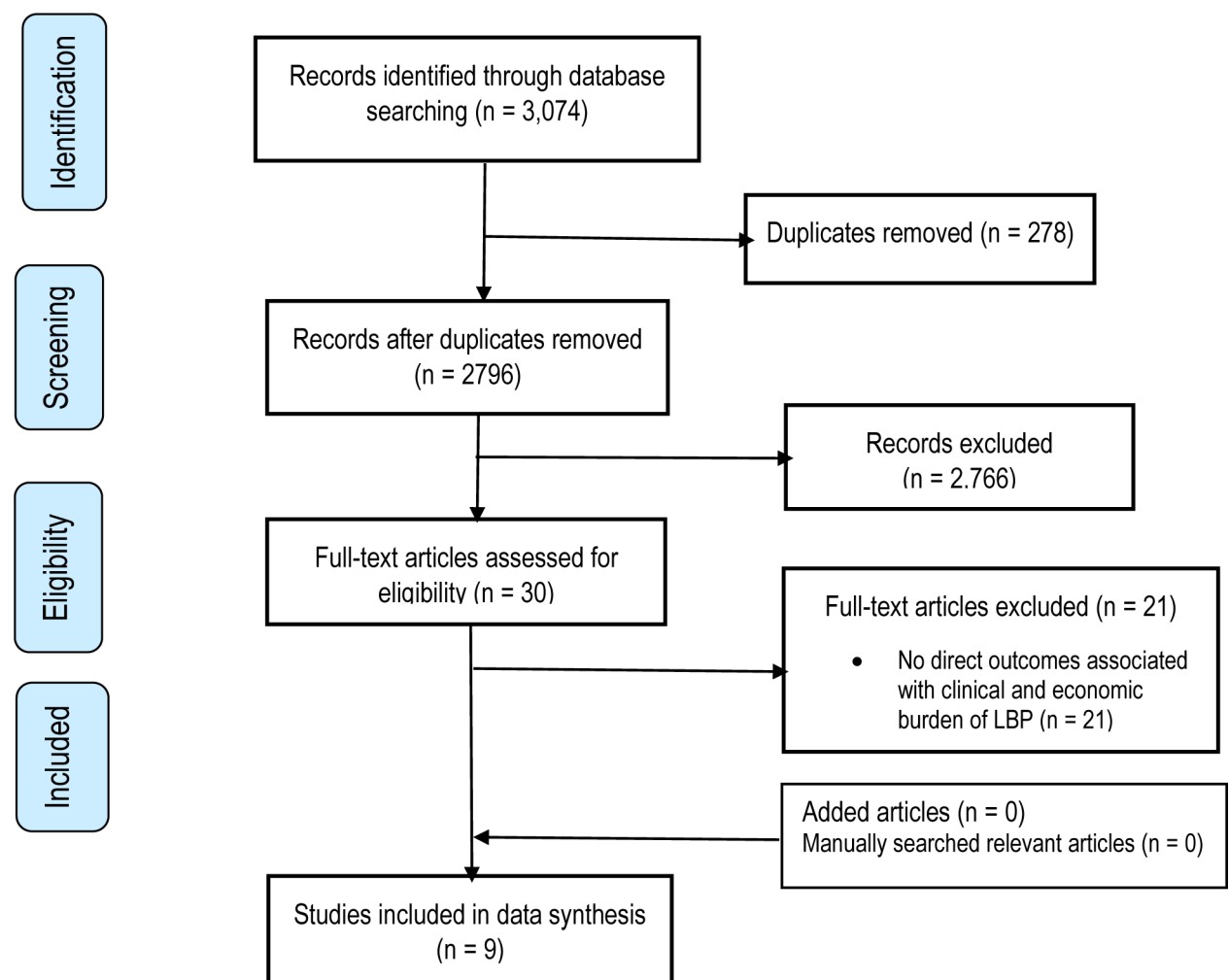

**Figure 1** Flow diagram of publications included and excluded in the review. LBP, low back pain.

**Table 1** Characteristics of clinical and economic burden studies included in the review

| Reference/country | Data source | Study objective | Inclusion criteria | Classification or diagnosis LBP | Conflict of interest | Funding source | NOS/9 |
|---|---|---|---|---|---|---|---|
| Beyera et al[16]/ Ethiopia | Interview | To analyse hospital admission and associated factors | Individuals with LBP of ≥18 years of age. Presented to healthcare facility for LBP in the past 1 year | Not provided | No | Not stated | 3 |
| Carregaro et al[17]/Brazil | National databases, considering the period 2012–2016. | To estimate the healthcare expenditure and productivity losses related to LBP | ≥19 years | ICD-10 codes | No | Yes | 6 |
| Laffont et al[18]/ Argentina | The Statistics and Information Department of the Ministry of Health | To analyse the impact of LBP on hospitalisations between 2006 and 2010. | Either adult LBP or lumbosciatica (≥18 years) | ICD-10 codes | No | No | 6 |
| Odole et al[19]/ Nigeria | Questionnaire | To determine the economic burden of LBP patients in Ibadan | Individuals with LBP receiving physiotherapy in Ibadan | Not provided | Not stated | Not stated | 3 |
| do Socorro Margarido et al[20]/Brazil | Questionnaire | To assess resource utilisation in the diagnosis, management and hospitalisation of patients | Participants were rheumatologists attending a national rheumatology medical congress. | Not provided | Not stated | No | 4 |
| Bello and Muhammad[22]/ Nigeria | Questionnaire | To estimate the cost of LBP management | Patients with LBP, allow them to carry out activities for daily living even under difficulties. | ICD-9 codes | No | No | 3 |
| Birabi et al[23]/ Nigeria | Medical records of hospitals (220 patients) | To report the results of a retrospective study on the direct cost burden of LBP between 2010 and 2013 | Patients whose care commenced from acute care stages, admitted where necessary. | ICD-10 codes | Not stated | Not stated | 5 |
| Lin et al[24]/ Republic of China | Interview | To estimate the 12-month economic cost of LBP among female (1990 to 1991) | Nurses who had experienced LBP within the 12-month period | ICD-10 codes | No | No | 3 |
| Radoičic et al[21]/ Republic of Serbia | Interview (April 2016 to April 2017). | To estimate the cost of LBP | Diagnosed as M54, which refers to lumbar pain (dorsalgia) | ICD-10 codes | No | Partially funded | 4 |

ICD, International Classification of Diseases; LBP, low back pain; NOS, Newcastle–Ottawa Quality Assessment Scale.

### Inclusion and exclusion criteria

Included in this review were studies among patients with LBP; original research findings related to costs (direct and indirect) in LMIC as defined by WHO; the contexts of interest were hospitals, primary healthcare clinics and home settings; observational (cross-sectional or surveys) published in peer-reviewed journals. Review articles, editorials, letters to the editor, news reports, conference abstracts, comments, languages other than English, as well as the results of dissertations were excluded. For ease

**Table 2** Definition of LBP in the included studies

| Study | Definition of LBP |
| --- | --- |
| Beyera et al[16] | Individuals who reported at least one presentation to any of the health institutions for their LBP in the past 1 year. |
| Carregaro et al[17] | ICD-10 codes were used to define the presence of LBP: M40.4 (other lordosis); M40.5 (lordosis, unspecified); M51 (other intervertebral disc disorders); M54.1 (radiculopathy); M54.3 (sciatica); M54.4 (lumbago with sciatica); M54.5 (low back pain); M54.8 (other dorsalgia) and M54.9 (dorsalgia, unspecified). |
| Laffont et al[18] | Not provided. |
| Odole et al[19] | Patients with mechanical LBP who were receiving physiotherapy on an outpatient. |
| do Socorro Margarido et al[20] | Not provided. |
| Bello and Muhammad[22] | Participants diagnosed and managed for LBP in secondary and tertiary hospitals for at least 1 year duration. The pain was low in the back and allow them to carry out activities for daily living even under difficulties. |
| Birabi et al[23] | Patients whose ages ranged between 26 and 65 years and were managed from acute care period for LBP (without discharge against medical advice) as well as were followed up on outpatient basis without default within the study period. |
| Lin et al[24] | The following definition of pain were used: minor: pain occurs when carrying 20 kg of weight for 10 min or more; (2) moderate: pain occurs when lifting 20 kg of weight; (3) serious: pain occur; when lifting less than 20 kg of weight; (4) severe: pain occurs when doing any task; (5) maximum: pain occurs even while at rest. |
| Radoičic et al[21] | Patients should have been diagnosed as M54, which refers to lumbar pain (dorsalgia). |

ICD, International Classification of Diseases; LBP, low back pain.

of interpretation, access and language barrier, reviews that were published or unpublished in another language were excluded.

## Study selection and assessment of methodological quality
After selection of the articles by the search strategy in each database, duplicate articles were excluded. Following the removal of duplicates, titles and abstracts were screened by two reviewers (TG and FF) to identify eligible studies. Articles were selected for inclusion based on predefined criteria like relevant patient population (LBP), appropriate study design and outcome measures (patient-level and population-level). Studies assessing the efficacy or effectiveness of specific interventions were not included.

Discrepancies for study selection were resolved through discussion with other investigators (CEM and UU). Having retrieved the full text of studies that met the inclusion criteria, they were assessed for methodological quality using the Newcastle–Ottawa Quality Assessment Scale (NOS) for cohort studies.[12] The NOS contains nine items, categorised into three dimensions including selection and comparability. Studies were scored using a scale with a possible maximum of nine points where a score ≥6 indicated high-quality studies, a score between 3 and 6 as moderate and a score ≤3 as low quality.

## Data extraction
Data were extracted by two independent review authors. The following information was extracted for each study: authors, country and year of publication, study objective, data source, inclusion criteria, LBP definition, population characteristics (size, % male and mean age), hospitalisation, average total annual cost per patient and annual population cost. Hospitalisation was defined as a primary discharge diagnosis of LBP, it could be obtained from hospital database records of discharge and admission date.[13] Summary table was used to display the extracted data. When there was disagreement, it was addressed through consultation with the third reviewer (CEM and UU).

## Data analysis
Given the heterogeneity of included studies in the review, it was not possible to undertake a meta-analysis. Instead, a narrative synthesis of the extracted data was employed to analyse, integrate and synthesise review findings. Study characteristics were first tabulated along with extracted outcomes to support the narrative synthesis. A preliminary synthesis was then undertaken by categorising extracted data clusters by reference/country, data source, study objective, inclusion criteria, LBP definition and clinical and economic burden of LBP. Using the tabulated data, relationships within and between included studies were then examined visually.

If necessary, costs were converted to US$ using salary converter.[14] Most of costs have been converted to US$ using purchasing power party and then inflation adjustments were calculated using the suggested country-specific inflation index from the World Bank Website.[15] Conversions were performed in February 2022. Further, hospitalisation rates were the number of patients with LBP hospitalised in a given period of follow-up. The higher rate of hospitalisation could mean substantial clinical burden of the condition.

## Patient and public involvement
Patients and the public were not involved in the design or planning of the study.

## RESULTS
This systematic review generated 3074 records in Scopus (n=51), PubMed (n=2036), Embase (n=700) and Medline,

**Table 3** Rate of hospitalisation for LBP patients

| Study | Number of participants | Year | Hospitalisations for LBP |
|---|---|---|---|
| Beyera et al[16] | 543 (M=316; F=205) | Between June and November 2018 | 14.4% (95% CI 11.4 to 17.3) with an average length of stay 7.4 days, 95% CI 6.4 to 8.8 |
| Carregaro et al[17] | 886 523 diagnostic imaging procedures for LBP | 2012 to 2016 | 13.4% (inpatient admissions) |
| Laffont et al[18] | A total of 154 430 discharges in which patients of any age were hospitalised | 2006 and 2010 | 18.7%, with an average length of stay in the hospital of 3.8 days |
| do Socorro Margarido et al[20] | 207 rheumatologists (M=62%; F=38%) | N/A | An average of 574 patients with LBP were seen per year. |

F, female; LBP, low back pain; M, male; N/A, not available.

AMED, CINAHL, PsycINFO (n=287) (figure 1). Of these, 278 were duplicates. Following screening by titles and abstracts, 2766 studies were excluded, leaving 30 articles for further full text review. No article was added through manual search of the bibliographies of relevant articles. After reading the full text, only nine met the inclusion criteria and were eligible for analysis. The list of the excluded studies was also reported (online supplemental appendix 2). The mean NOS score of the included studies was 4, with an average from 3 to 6 (table 1). The NOS score showed that all the included studies do not show comparability of cohorts on the basis of age, sex, the design and analysis.

## Characteristics of the included studies
The characteristics of the included studies are presented in table 1. The clinical and economic burden of LBP studies were conducted in Nigeria (n=3), Ethiopia (n=1), Brazil (n=2), China (n=1), Republic of Serbia (n=1) and Argentina (n=1). Of the nine eligible studies included in our review, three studies[16–18] reported the clinical impact of LBP in LMICs. Whereas the remaining studies reported the economic burden of LBP. The data sources of the included studies were interviews, questionnaire, population-based cross-sectional study and medical records. Except three studies,[16 19 20] all the included studies reported LBP classification or diagnosis.

## Definition of LBP
The definition of LBP adopted by the included studies is presented in table 2. Except two studies,[18 20] all the included studies provided the definition of LBP. The definition adopted by each included study is not the same. For example, two studies[17 21] used ICD-codes to define LBP. Other study[16] adopted the frequency of LBP report to health institution.

**Table 4** Summary of studies that reported patient-level and population-level total costs for LBP

| Reference | Age group (years) | Average annual cost per patient | Inflated 2022 $US |
|---|---|---|---|
| Carregaro et al[17] | ≥19 | Total costs=US$ 2.2 billion* Indirect cost=1.7 billion* (productivity losses) | Total cost=US$2.5 billion* |
| Odole et al[19] | 30–80 | Total cost=US$1125.27±623.39 Direct costs=US$927.20±600.26. Indirect cost=US$198.08±136.15 | Total cost=US$5047.5±2796.3 Direct costs=US$4158.1±2691); Indirect cost=US$888±610. |
| do Socorro Margarido et al[20] | N/A | 30% of patients with LBP would be absent from their work due to the condition. | N/A |
| Bello and Muhammad[22] | 30–70 | Total=US$763.27±543.97; Direct cost=US$647.28±438.99; Indirect cost=US$115.99±104.99 | Total=US$ 3029.9±2159.1 Direct cost=US$2569.5±1742.6 Indirect cost=460.1 ± 416.5 |
| Birabi et al[23] | 26 and 65 | Total cost=US$1226.25 (government hospitals); US$4884.37 (private hospitals) | #US$3192.53 (government hospitals) #US$12 716.41 (private hospitals) |
| Lin et al[24] | Mean age=32.2±9.2 | Total costs=US$994.4 to US$1406.4. | N/A |
| Radoičic et al[21] | Median, 53.59 (28 – 85) | Total costs = €200.40 ± €86.65, Direct costs = €9.39 ± €6.66. Indirect costs = €182.00 ± €78.66 | Total cost = €432.86 ± €187.2 Total indirect costs = €393.12 ± €169.99 |

*Per population.
LBP, low back pain; N/A, not available.

## Hospitalisation

A total of three studies reported data on rates of hospitalisation for LBP (table 3). The rates of hospitalisation ranged between 13.4% and 18.7%. The highest rate of hospitalisation (18.7%) for LBP was reported in Argentina.[18]

## Economic burden

The economic burden of the included studies is provided in table 4. A total of six studies reported average annual cost per patient.[16 19 21–24] One of the included studies reported annual cost per population.[18] Patients with the age range up to 85 years were included in the included studies. Due to variation in methodological approaches, the reported cost estimates were inconsistent across the studies. However, up to an average of annual costs of US$2.2 billion and US$1226.25 per population and per patient level, respectively, were reported. Up to US$1.7 billion indirect costs were also reported mainly due to work absenteeism.[18]

## DISCUSSION

To our knowledge, this is the first systematic review to analyse the clinical and economic impact of LBP in LMICs. A total of nine studies from six LMICs were included. Our review indicates that the clinical and economic burden of LBP is substantial in LMICs. The rate of hospitalisation ranged between 13.4% and 18.7% of patients with LBP. The findings of the review also suggest that LBP is associated with an annual cost of US$2.2 billion per population in LMICs. A clearer understanding of the clinical and economic burden of LBP in LMICs is therefore important for policy makers that they may consider appropriate prevention and management strategies for LBP.

Rate of hospitalisation of female patients with LBP was higher (53.7%) than their male counterpart.[20] The reason for this could be epidemiological where the proportion of LBP in female is in a higher proportion than males.[25] The average length of stay in hospital due to LBP in Argentina and Ethiopia was 3.8 days[20] and 7.4 days,[17] respectively. The length of stay in hospital is directly associated with high financial impact of LBP, this is because of its healthcare and social costs.[26]

We collected evidence of both direct and indirect economic burden of LBP. The overall burden varied based on characteristics of patient populations, geographic region and the methodologies adopted by the studies, this includes the individual patient or the population-level perspective. For example, the societal annual costs amounted to US$ 2.2 billion per population where the productivity losses represented 79% of the costs reported in Brazil.[18] Overall, the considerable clinical and economic burden of LBP in LMICs could be due to its poorly equipped health and social system.[27]

There are certain strength and limitations to this study. The main strength of this review is the comprehensiveness of the search terms, screening of numerous data bases and assessment of methodological quality of the studies. Only English language studies were included. Therefore, it is possible that relevant literature published in other languages may have been excluded. Also, grey literature were not included in the current review, as a result, we cannot be certain that the findings of this review are representative of all the studies undertaken, negative results are less likely to be published which may result in publication bias. The quality of the included studies was evaluated using NOS as moderate; however, the instrument used has some limitations as important methodological issues pertaining to health economics might not have been fully assessed by scale. It was not possible to undertake a meta-analysis for most of the included studies due to the adoption of different time horizons, different clinical outcomes and lack of data regarding mean and SD for costs.[28] Further, the quality of the included studies indicates that there is a need to interpret the clinical and economic burden results with caution.

## CONCLUSION

We found that LBP is associated with significantly high rate of hospitalisation with a clear impact on length of stay in hospital and treatment costs. The findings of our analysis can be used by clinicians and policymakers to better allocate resources for prevention and management strategies for LBP to improve health outcomes and reduce the substantial burden associated with the condition. We also hope our results will be of use to researchers planning to evaluate the cost-effectiveness of various strategies for preventing LBP in LMICs.

**Contributors** FF and TG performed electronic searches and assessed title and abstract. CEM and UU were consulted in case of disagreement. FF and TG performed the methodological quality assessment and extracted the data. TG and CEM wrote the original draft preparation. FF and UU reviewed, revised, and edited this paper. All authors read and approved the final draft of the manuscript. FF had full access to all of the data in the study, and takes responsibility for the integrity of the data and the accuracy of the data analysis. FF is the guarantor.

**Funding** The authors have not declared a specific grant for this research from any funding agency in the public, commercial or not-for-profit sectors.

**Competing interests** None declared.

**Patient and public involvement** Patients and/or the public were not involved in the design, or conduct, or reporting, or dissemination plans of this research.

**Patient consent for publication** Not applicable.

**Ethics approval** Not applicable.

**Provenance and peer review** Not commissioned; externally peer reviewed.

**Data availability statement** All data relevant to the study are included in the article or uploaded as supplementary information. All data related to this work are available.

**ORCID iD**
Francis Fatoye http://orcid.org/0000-0001-7976-2013

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
