## [Reviewer comments · BMJ Open]

ARTICLE DETAILS

TITLE (PROVISIONAL)	Clinical and economic burden of low back pain in low- and middle-income countries: a systematic review
AUTHORS	Fatoye, Francis; Gebrye, Tadesse; Mbada, Chidozie Emmanuel; Useh, Ushotanefe

VERSION 1 – REVIEW

REVIEWER	Mbuzeleni Hlongwa University of KwaZulu-Natal College of Health Sciences, Nursing and Public Health
REVIEW RETURNED	30-Jun-2022

GENERAL COMMENTS	Thank you for the opportunity to review the manuscript. I found the manuscript to be well written and it addresses an important public health issue. I have made minor comments below: • “Further, a manual search of reference sections of the included studies was also checked for additional studies” What was the outcome of this manual search, did it yield any additional articles?• Page 7, “Most of costs have been converted to US\$ using purchasing power parity and then inflation adjustments were calculated using the suggested country specific inflation index at” this sentence seems incomplete• “A narrative synthesis of the extracted data was employed” please provide a step-by-step process which was followed here?• Why were sensitivity analyses not conducted in this review?• Did the authors search the Cochrane library to check whether there is a recently published or ongoing review in the same topic?o Please indicate the differences between this review to this one: https://www.mdpi.com/1660-4601/19/5/2964• The authors indicate that quality appraisal was conducted but did not elaborate further on their findings, unless I missed this – which studies performed low, on what aspects, and what does this mean for this review? Such information is necessary
---

REVIEWER	Rodrigo Caregarro Universidade de Brasília, Health Sciences
REVIEW RETURNED	26-Sep-2022

GENERAL COMMENTS	Thanks for the opportunity of revising this manuscript. The topic is relevant for decision-makers and health professionals. However, I have several concerns and specific methodological comments that might help to improve this manuscript. In addition, I've checked the protocol registration (Prospero) and found numerous discrepancies. The authors should be advised that the protocol is relevant for the review and research transparency. Finally, I have some concerns
---

	regarding the critical appraisal, as you did not adopt a proper instrument designed for health economic studies. Please check my comments below. General comments:  - The register within PROSPERO indicates that the review has not started, although the authors submitted the manuscript. Therefore, there are some issues pertaining to the update of the protocol registration. Specific comments:  - Page 5, search strategy: If the aim is to summarize evidence from low-middle-income countries, why did you limit the search to the English language? This must be justified, as important studies might have been not found or excluded (for instance, from Spanish or Portuguese-language countries). - Why did you not search for studies within the grey literature? Please clarify. - Please include a review question and clarify if you adopted some sort of acronym used to support the search strategy elaboration. - Please include an appendix with the actual search strategies used within the major databases (Medline, Embase, Scopus). This might help to improve the study reproducibility and also the transparency of the authors strategy. This is relevant because I performed a quick search and found at least 1 study dealing with spinal disorders (but also investigated low back pain), and was not found by the search strategy (or was excluded, hence, my next comment): https://link.springer.com/article/10.1007/s00038-019-01211-6 - Please include an appendix table including the reference (author, year), country, aims, design, and justification for the exclusion of the 21 studies during the full-text reading. This is mandatory to improve the transparency of the review and also to guarantee that there was no selection bias. - Regarding data extraction: I strongly recommend that you include, within the studies characteristics (table 1), information pertaining conflict of interests and funding source. This is relevant and might also improve the critical analysis of the included studies. - The authors adopted the NOS scale for methodological quality/bias, however, in page 5, lines 23-24 they stated that RCTs and modelling studies could also be included. In addition, within the PROSPERO register the authors state that: "The study designs that will be included for the review are studies conducted based on the prevalence and incidence based approach". This is somewhat confusing and conflicting, because the NOS scale was designed for observational studies. The criteria within the register are to broaden and the authors considered a more specific inclusion criteria within the manuscript. From my standpoint, this is a protocol deviation. - Moreover, the NOS scale is not fully adequate to assess the methodological quality of health economic studies. Therefore, this is a weakness/limitation of the current review. I strongly recommend that the authors include a second appraisal instrument, in order to adequately investigate the quality of this study design. For instance, there is no appraisal regarding the perspective, approach and methods pertaining cost of illness studies (e.g., those performing macrocosting or microcosting analysis), type of costs, if resources were valued appropriately, and so on. I would recommend the reading of the following critical guide: https://pubmed.ncbi.nlm.nih.gov/21604822/ - Curiously, the authors considered ICD-10 codes as proper definitions of the case (low back pain). I disagree and, instead, I suggest that the current review must provide a proper definition
--	---

	(synthesized from the literature). Subsequently, this definition must be critically appraised in the included studies to check if a sound and clinically relevant definition was adopted. If I'm not mistaken, several of the included studies investigated national databases or system surveys, hence, ICD-10 codes are used within these systems as classification or diagnosis. However, this does not mean that the studies considered a proper definition of the condition (low back pain) and reported how they decided to choose the specific ICD-10 codes related to low back pain. Please correct accordingly. - Page 6: Why you presented separate search findings from PubMed and Medline? My impression of the method was that the search was performed at Medline via Pubmed. Hence, why two separate search findings? This is rather confusing. Also, AMED was not described within the protocol registration, and Cochrane database was not described in the manuscript, although it was included in the protocol (quote from PROSPERO: "We will conduct a systematic search on PubMed, MEDLINE, EMBASE, Scopus, PsycINFO, Cochrane Database of Systemic Reviews). - Table 2: I am struggling to understand how 880,000 diagnostic images are considered the number of participants from the reference n. 16? I strongly recommend including what you considered as the rate of hospitalization within your method section. Accordingly, please include a data analysis section and explain your variables/analysis.
--	---

VERSION 1 – AUTHOR RESPONSE

Reviewer 1	
"Further, a manual search of reference sections of the included studies was also checked for additional studies" What was the outcome of this manual search, did it yield any additional articles?	This has now been addressed (see the result section, p. 1)
Page 7, "Most of costs have been converted to US\$ using purchasing power parity and then inflation adjustments were calculated using the suggested country specific inflation index at" this sentence seems incomplete	This has now been addressed (see the method section, data synthesis)
"A narrative synthesis of the extracted data was employed" please provide a step-by-step process which was followed here?	This has now been addressed (see the method section, data synthesis)
Why were sensitivity analyses not conducted in this review?	Due to the heterogeneity of the included studies we were unable to combine the statistical data and perform sensitivity analysis.
Did the authors search the Cochrane library to check whether there is a recently published or ongoing review in the same topic?	We checked if there is published or ongoing review on both PROSPERO and the Cochrane library and we found no reviews with the same topic.

Please indicate the differences between this review to this one: https://www.mdpi.com/1660-4601/19/5/2964	The current review is on clinical and economic burden of low back pain in low- and middle-income countries whereas this is: https://www.mdpi.com/1660-4601/19/5/2964, on the epidemiology of chronic low back pain among adults in Sub-Saharan Africa.
The authors indicate that quality appraisal was conducted but did not elaborate further on their findings, unless I missed this – which studies performed low, on what aspects, and what does this mean for this review? Such information is necessary	The quality assessment of the included studies was placed in Table 1 (NOS/9) and described on the result section (1st paragraph)
Reviewer 2	
I've checked the protocol registration (Prospero) and found numerous discrepancies. The authors should be advised that the protocol is relevant for the review and research transparency. Finally, I have some concerns regarding the critical appraisal, as you did not adopt a proper instrument designed for health economic studies. Please check my comments below. General comments: The register within PROSPERO indicates that the review has not started, although the authors submitted the manuscript. Therefore, there are some issues pertaining to the update of the protocol registration.	We have updated the protocol registration.
Page 5, search strategy: If the aim is to summarize evidence from low-middle-income countries, why did you limit the search to the English language? This must be justified, as important studies might have been not found or excluded (for instance, from Spanish or Portuguese-language countries).	We believe that there is little evidence of a systematic bias from the use of language restrictions in systematic review (https://pubmed.ncbi.nlm.nih.gov/22559755/). Further, we have mentioned the impact of language restrictions in the discussion part of the review (paragraph 4).
Why did you not search for studies within the grey literature? Please clarify.	Grey literature was excluded due to our inclusion and exclusion criteria. Grey literature includes studies that are not peer-reviewed and unpublished literature.
Please include a review question and clarify if you adopted some sort of acronym used to support the search strategy elaboration.	This has now been addressed, please see the method section (Study selection and assessment of methodological quality)
Please include an appendix with the actual search strategies used within the major databases (Medline, Embase, Scopus). This might help to improve the study reproducibility and also the transparency of the authors strategy. This is relevant because I performed a quick search and found at least 1 study dealing with spinal disorders (but also	We have included the search strategy in Appendix. The study dealing with the spinal disorders was not considered for our review, there is no clinical or economic burden of low back pain.

investigated low back pain), and was not found by the search strategy (or was excluded, hence, my next comment): https://link.springer.com/article/10.1007/s00038-019-01211-6	
Please include an appendix table including the reference (author, year), country, aims, design, and justification for the exclusion of the 21 studies during the full-text reading. This is mandatory to improve the transparency of the review and also to guarantee that there was no selection bias.	We have now reported the excluded studies.
Regarding data extraction: I strongly recommend that you include, within the studies characteristics (table 1), information pertaining conflict of interests and funding source. This is relevant and might also improve the critical analysis of the included studies.	This has now been addressed.
The authors adopted the NOS scale for methodological quality/bias, however, in page 5, lines 23-24 they stated that RCTs and modelling studies could also be included. In addition, within the PROSPERO register the authors state that: "The study designs that will be included for the review are studies conducted based on the prevalence and incidence based approach". This is somewhat confusing and conflicting, because the NOS scale was designed for observational studies. The criteria within the register are to broaden and the authors considered a more specific inclusion criteria within the manuscript. From my standpoint, this is a protocol deviation.	Correction was made regarding the inclusion criteria (randomised controlled trials (RCTs) and modelling studies).
Moreover, the NOS scale is not fully adequate to assess the methodological quality of health economic studies. Therefore, this is a weakness/limitation of the current review. I strongly recommend that the authors include a second appraisal instrument, in order to adequately investigate the quality of this study design. For instance, there is no appraisal regarding the perspective, approach and methods pertaining cost of illness studies (e.g., those performing macrocosting or microcosting analysis), type of costs, if resources were valued appropriately, and so on. I would recommend the reading of the following critical guide: https://pubmed.ncbi.nlm.nih.gov/216048	We adopted the NOS scale to assess the methodological quality of the included studies. Many reviews of the same topic have used the same tool. # Ho, S. C., Chong, H. Y., Chaiyakunapruk, N., Tangiisuran, B., & Jacob, S. A. (2016). Clinical and economic impact of non-adherence to antidepressants in major depressive disorder: a systematic review. Journal of affective disorders, 193, 1-10. # van Boven, J. F., Chavannes, N. H., van der Molen, T., Rutten-van Mölken, M. P., Postma, M. J., & Vegter, S. (2014). Clinical and economic impact of non-adherence in COPD: a systematic review. Respiratory medicine, 108(1), 103-113.

22/	
Curiously, the authors considered ICD-10 codes as proper definitions of the case (low back pain). I disagree and, instead, I suggest that the current review must provide a proper definition (synthesized from the literature). Subsequently, this definition must be critically appraised in the included studies to check if a sound and clinically relevant definition was adopted. If I'm not mistaken, several of the included studies investigated national databases or system surveys, hence, ICD-10 codes are used within these systems as classification or diagnosis. However, this does not mean that the studies considered a proper definition of the condition (low back pain) and reported how they decided to choose the specific ICD-10 codes related to low back pain. Please correct accordingly.	The case definition of LBP differed considerably among the studies. Some of the studies specified that algorithms were used to estimate the probability of medical service utilization being related to LBP based on 66 ICD-9 codes, whereas others did not provide any operational definition of LBP. #Dagenais, S., Caro, J., & Haldeman, S. (2008). A systematic review of low back pain cost of illness studies in the United States and internationally. The spine journal, 8(1), 8-20. # This has now been addressed.
Page 6: Why you presented separate search findings from PubMed and Medline? My impression of the method was that the search was performed at Medline via Pubmed. Hence, why two separate search findings? This is rather confusing. Also, AMED was not described within the protocol registration, and Cochrane database was not described in the manuscript, although it was included in the protocol (quote from PROSPERO: "We will conduct a systematic search on PubMed, MEDLINE, EMBASE, Scopus, PsycINFO, Cochrane Database of Systemic Reviews).	We can access Medline, Cinhal, PsycINFO and AMED from our Manchester Metropolitan University simultaneously (https://web-s-ebSCOhost-com.mmu.idm.oclc.org/ehost/search/advanced?vid=0&sid=a5ca9c13-6f0d-4fb0-aedd-3fd18fd29111%40redis). Yes, we also have made some modification to the protocol registration to improve the outcome of the search strategy.
Table 2: I am struggling to understand how 880,000 diagnostic images are considered the number of participants from the reference n. 16? I strongly recommend including what you considered as the rate of hospitalization within your method section. Accordingly, please include a data analysis section and explain your variables/analysis.	This has been addressed, please see the method section (data extraction & data analysis).

VERSION 2 – REVIEW

REVIEWER	Rodrigo Caregarro Universidade de Brasília, Health Sciences
REVIEW RETURNED	21-Jan-2023
GENERAL COMMENTS	Dear authors, thanks for addressing my previous comments. The current version is improved and well-written. I have only minor

	comments at this time:  - Regarding the grey literature: I maintain my recommendation to include data from the grey literature. This is relevant for evidence synthesis, in order to avoid the risk of publication bias. Even if the authors choose not to include this kind of information, it is mandatory to acknowledge that there is a risk of publication bias. By not including the grey literature, you cannot guarantee that your evidence synthesis covered all evidence available, and the reader may assume that important studies might have not been included in the systematic review. - Regarding the NOS scale, I strongly suggest that you include a limitation addressing the fact that this instrument has limitations to assess health economics studies. Even though you presented previous studies using the NOS scale, this does not mean that they are correct. Hence, a limitation would be adequate to recognize that important methodological issues pertaining to health economics might not have been adequately assessed.
--	---

VERSION 2 – AUTHOR RESPONSE

Reviewer 2: Dr. Rodrigo Caregarro	
Regarding the grey literature: I maintain my recommendation to include data from the grey literature. This is relevant for evidence synthesis, in order to avoid the risk of publication bias. Even if the authors choose not to include this kind of information, it is mandatory to acknowledge that there is a risk of publication bias. By not including the grey literature, you cannot guarantee that your evidence synthesis covered all evidence available, and the reader may assume that important studies might have not been included in the systematic review.	This has now been addressed (see the discussion (limitation) section, p. 11)
Regarding the NOS scale, I strongly suggest that you include a limitation addressing the fact that this instrument has limitations to assess health economics studies. Even though you presented previous studies using the NOS scale, this does not mean that they are correct. Hence, a limitation would be adequate to recognize that important methodological issues pertaining to health economics might not have been adequately assessed.	This has now been addressed (see the discussion (limitation) section, p. 11)